# Ilimaquinone Induces the Apoptotic Cell Death of Cancer Cells by Reducing Pyruvate Dehydrogenase Kinase 1 Activity

**DOI:** 10.3390/ijms21176021

**Published:** 2020-08-21

**Authors:** Choong-Hwan Kwak, Ling Jin, Jung Ho Han, Chang Woo Han, Eonmi Kim, MyoungLae Cho, Tae-Wook Chung, Sung-Jin Bae, Se Bok Jang, Ki-Tae Ha

**Affiliations:** 1Korean Medical Research Center for Healthy Aging School of Korean Medicine, Pusan National University, Yangsan, Gyeongnam 50612, Korea; hahaaaa@nate.com (C.-H.K.); twchung@pusan.ac.kr (T.-W.C.); Dr.NowOrNever@pusan.ac.kr (S.-J.B.); 2Department of Korean Medical Science, School of Korean Medicine, Pusan National University, Yangsan, Gyeongnam 50612, Korea; jinling0122@pusan.ac.kr (L.J.); hanjh1013@pusan.ac.kr (J.H.H.); 3Department of Molecular Biology, College of Natural Sciences, Pusan National University, Geumjung-gu, Busan 46241, Korea; hotorses@naver.com (C.W.H.); sbjang@pusan.ac.kr (S.B.J.); 4National Institute for Korean Medicine Development, Gyeongsan, Gyeongbuk 38540, Korea; minnie60@nikom.or.kr (E.K.); meanglae@nikom.or.kr (M.C.)

**Keywords:** ilimaquinone, Warburg effect, pyruvate dehydrogenase kinase 1, glycolysis, apoptosis

## Abstract

In cancer cells, aerobic glycolysis rather than oxidative phosphorylation (OxPhos) is generally preferred for the production of ATP. In many cancers, highly expressed pyruvate dehydrogenase kinase 1 (PDK1) reduces the activity of pyruvate dehydrogenase (PDH) by inducing the phosphorylation of its E1α subunit (PDHA1) and subsequently, shifts the energy metabolism from OxPhos to aerobic glycolysis. Thus, PDK1 has been regarded as a target for anticancer treatment. Here, we report that ilimaquinone (IQ), a sesquiterpene quinone isolated from the marine sponge *Smenospongia cerebriformis*, might be a novel PDK1 inhibitor. IQ decreased the cell viability of human and murine cancer cells, such as A549, DLD-1, RKO, and LLC cells. The phosphorylation of PDHA1, the substrate of PDK1, was reduced by IQ in the A549 cells. IQ decreased the levels of secretory lactate and increased oxygen consumption. The anticancer effect of IQ was markedly reduced in PDHA1-knockout cells. Computational simulation and biochemical assay revealed that IQ interfered with the ATP binding pocket of PDK1 without affecting the interaction of PDK1 and the E2 subunit of the PDH complex. In addition, similar to other pyruvate dehydrogenase kinase inhibitors, IQ induced the generation of mitochondrial reactive oxygen species (ROS) and depolarized the mitochondrial membrane potential in the A549 cells. The apoptotic cell death induced by IQ treatment was rescued in the presence of MitoTEMPO, a mitochondrial ROS inhibitor. In conclusion, we suggest that IQ might be a novel candidate for anticancer therapeutics that act via the inhibition of PDK1 activity.

## 1. Introduction

It is known that, under oxygen-abundant conditions, glycolysis rather than oxidative phosphorylation (OxPhos) is normally preferred as the energy source in cancer cells; this phenomenon is termed the Warburg effect [1]. Pyruvate dehydrogenase kinases (PDKs) are the primary metabolic enzymes involved in the Warburg effect. Four different isoforms of PDK have been found in mammals, and these enzymes regulate the activity of the pyruvate dehydrogenase complex (PDC) by phosphorylating its E1 alpha subunit (PDHA1) [2]. The PDC is divided into mainly three parts: pyruvate dehydrogenase E1 subunit, dihydrolipoamide acetyltransferase (the E2 subunit), and lipoamide dehydrogenase (the E3 subunit); PDKs bind to the E2 subunit and phosphorylate PDHA1 [3]. The PDC converts pyruvate into acetyl-CoA, which enters the tricarboxylic acid cycle; thus, the activity of the PDC is very important for OxPhos.

It is well known that, in many types of cancers, PDKs, especially PDK1, are overexpressed and/or its activity is increased by post-translational modifications [4,5,6]. In addition, PDK1 is highly expressed in non-small cell lung cancer (NSCLC), and its overexpression is significantly correlated with poor prognosis in human NSCLC tissues [7]. Furthermore, levels of PDK1 and phosphor-PDHA1 are regarded as prominent negative prognostic factors in NSCLC [8]. Pyruvate could be converted to lactate by lactate dehydrogenase A (LDHA) when the activity of the PDH complex was reduced by PDK1 activation [9]. The reprogramming of cancer metabolism through the enhancement of PDC activity via the inhibition of PDK1 increases the anticancer effects on several cancer cell lines, including NSCLC [10,11]. Thus, our group has continued the efforts to discover novel PDK1 inhibitors from natural compounds as candidates for anticancer drugs [2,12,13].

Marine sponges contain many secondary metabolites with potential biological benefits. Ilimaquinone (IQ) is a sesquiterpene quinone isolated from the marine sponge *Smenospongia cerebriformis* Duchassaing and Michelotti [14]. IQ has various biological functions, such as anti-HIV, antimicrobial, anti-inflammatory, and anticancer effects [15]. IQ has been reported to exert its anticancer activity via a GADD153-mediated pathway in prostate cancer cells and promote TRAIL-induced apoptosis via ROS-ERK/p38 MAPK-CHOP pathways in human colon cancer cells [15,16]. Moreover, IQ induces apoptosis in human colon cancer cells via p53 activation [17]. However, the anticancer mechanisms of IQ via PDK1 inhibition have not yet been reported.

In this study, we revealed that IQ isolated from *S. cerebriformis* induced the apoptosis of lung cancer cells by inhibiting the activity of PDK1. To the best of our knowledge, this is the first report on the anticancer effect of IQ via the inhibition of PDK1 activity.

## 2. Results

### 2.1. IQ Decreases Cancer Cell Viability and Inhibits PDK Activity

Firstly, the cytotoxicity of IQ on several lung and colorectal cancer cell lines was examined. The viability of each cell line was significantly decreased by IQ treatment (Figure 1). The 50% growth inhibition (GI_50_) values of IQ were shown in Table 1. Because the GI_50_ value of IQ in the A549 lung cancer cells was the lowest, further investigation was performed using this cell line. In addition, cell viability was markedly reduced by IQ treatment when compared with that by JX06 and AZD8545, well-known PDK inhibitors (Appendix A). Next, the inhibitory effect of IQ on the activity of PDK1 was estimated via Western blotting using a phosphor-PDHA1 antibody. The phosphorylation of PDHA1 was markedly reduced by IQ treatment up to 50 μM for 4 h (Figure 2A). In this condition, the survival of A549 cells did not show a significant reduction (data not shown). To rule out whether the reduction in PDHA1 phosphorylation was due to PDKs expression, the protein levels of PDKs were examined. IQ did not affect the expression of PDK1–4 in the conditions (Figure 2B). These data suggested that IQ decreased PDHA1 phosphorylation through suppressing the activity of PDKs but did not affect their expression. Furthermore, because PDHA1 activation promotes OxPhos, we examined the O_2_ consumption and secreted lactate levels. IQ decreased lactate production and increased O_2_ consumption in the A549 cells (Figure 2C,D). To ensure whether the inhibition of PDK activity is directly correlated with the cytotoxic effect of IQ, we constructed PDHA1-knockout A549 cells using a clustered regularly interspaced short palindromic repeats (CRISPR) system (Appendix A). The proliferation of PDHA1-knockout cells did not show a significant difference compared with parent A549 cells (data not shown). However, the cytotoxicity of IQ was significantly recovered in the PDHA1-knockout cells (Figure 2E). From these results, we supposed that IQ could increase PDH activity and OxPhos, and subsequently, reduce cell viability via suppressing PDKs activity.

### 2.2. IQ Inhibits PDK1 Enzyme Activity by Interfering with ATP Binding

The binding affinity of IQ to PDK1 was determined via auto-isothermal titration calorimetry (Auto-ITC) (Figure 3A). IQ showed high binding affinity to PDK1, with an equilibrium dissociation constant (Kd) of 14.64 μM. To elucidate the inhibitory mechanism of IQ on PDK activity, we generated a structural model of PDK1 and IQ interaction (Figure 3B). IQ was predicted to bind to the polar residue (Asn283) and the positive charge residue (Arg286) of PDK1. The residues are near the adenosine triphosphate (ATP)-binding domain at the C-terminus of PDK1, which is located around the two helices (α10 and α11) and the loop between α11 and α12 [18]. In addition, IQ bound to a similar site in the ATP binding pocket of the PDK isoforms (Appendix A). Furthermore, the (α-^32^P) ATP binding of PDK1 was decreased in the presence of IQ (Figure 3C). However, IQ did not affect binding between PDK1 and the PDH-E2 subunit (Figure 3D).

### 2.3. IQ Induces Mitochondrial ROS-Dependent Apoptosis in the A549 Cells

It was reported that the inhibition of PDK1 activity induces the production of mitochondrial ROS in cancer cells [19,20]. As in previous studies, the mitochondrial ROS was significantly increased in the presence of IQ (Figure 4A,B). Moreover, MitoTEMPO, a mitochondrial ROS inhibitor, significantly decreased the IQ-induced increase in mitochondrial ROS production (Figure 4A,B). When mitochondrial ROS increased, the membrane potential of mitochondria consequently decreased [2]. Thus, we examined the effect of IQ on the membrane potential of mitochondria using TMRM staining. The mitochondrial membrane potential was also markedly depolarized by IQ treatment using confocal microscope imaging (Figure 4C) and FACS analysis (Figure 4D,E). These results suggested that IQ significantly induces the mitochondrial damage by inhibiting the PDK1 activity.

To confirm whether the type of cell death induced by IQ was apoptosis, we examined annexin V-propidium iodide (PI) double staining. As shown Figure 5A,B, IQ significantly induced the population of annexin V-PI-positive cells. Furthermore, as determined by Western blot analysis, IQ significantly induced cleavage of caspase 3, -9, and PARP in A549 cells (Figure 5C). These data suggested that IQ induces apoptosis in A549 cells. To confirm the mitochondrial ROS-dependent apoptosis, we examined cell viability and annexin V-PI staining in the presence of MitoTEMPO and IQ. The pretreatment with MitoTEMPO reduced the cytotoxic effect of IQ, and the population of apoptotic cells was clearly rescued (Figure 5D–F). These results suggested that IQ induces the mitochondrial ROS-dependent apoptosis in A549 cells.

## 3. Discussion

In this study, we showed that IQ suppressed the growth of several types of cancer cells by inhibiting the kinase activity of PDK1 by interfering with ATP binding, subsequently dephosphorylating and activating PDHA1. The activation of PDH activity upon IQ treatment reduced lactate production and accelerated OxPhos, thereby increasing the ROS production and membrane depolarization of the mitochondria in the A459 lung cancer cells. The IQ-induced mitochondrial damage ultimately resulted in increased apoptotic cell death (Appendix A). These results showed a good correlation with previous findings that the genetic or pharmacological suppression of PDK1 induces electron transport chain complex 1-based mitochondrial ROS production by increasing levels of NADH [19,21]. Under these conditions, electron transport chain complex 1 is damaged by the mitochondrial ROS, which consequently leads to the release of pro-apoptotic factors [6,19]. To confirm whether the anticancer effect of IQ was related to inhibition of PDK1 activity, we used PDHA1-knockout A549 cells. In the PDHA1-knockout cells, cell viability was partially rescued by about 40% compared with IQ-treated wild type A549 cells. The results might due to off-targets of IQ or another target of PDK1, including EGFR, HGF-MET, and PARL [22,23,24]. Thus, we suggest that IQ significantly induced mitochondrial ROS production by inhibiting PDK1 activity and thereby induced apoptotic cell death in the A549 lung cancer cells.

IQ is a chemical constituent in the genus *Spongia*, including *Hippiospongia metachromia* and *S. cerebriformis* [25,26]. The antimicrobial, anticancer, anti-inflammatory, and protein transport-inhibitory effects of IQ have been reported [14,16,27,28]. The pharmacokinetic features of IQ were also previously studied in rat models [29,30]. In a previous study, the GI_50_ values of IQ on PC-3, DU145, LNCaP, MG63, A549, and Hep3B cells were found to be 2.6 to 19.8 μM at 48 h [16]. The GI_50_ values of IQ following 24-h treatment, as confirmed in this study, were slightly higher than those reported in the previous study, but the difference was slight. Regarding the molecular mechanism underlying the IQ-induced apoptosis, CHOP/GADD153-mediated cell death as well as p53 activation-mediated apoptosis and autophagy were reported [15,16,17]. It was reported that the ROS-independent ER stress pathway, which is mediated by GRP78/p-ERK/NRF2 signaling, activates PDK1 activity, and mediates the metabolic shift toward the Warburg effect in cancer-initiating cells [31]. In addition, increased p53 activity negatively regulates PDK2 expression [32]. These studies suggest that complex molecular mechanisms might be related in IQ-induced cell death, such as PDK1 activity, ER stress, and p53 activity. To elucidate the precise mechanism underlying the anticancer mechanism of IQ with regard to ER stress, autophagy, and p53 regulation, extensive experiments are required.

There are four different binding sites in small molecule inhibitors that can be targeted to suppress PDK activity, including the nucleotide (ATP)-binding site, pyruvate-binding site, lipoamide-binding site, and allosteric CoA-binding site [33]. Several small molecules, such as radicicol, M77976, and hordenine, are known to be PDK inhibitors that interfere with ATP binding [18,33,34]. Recently, hordenine, an alkaloid isolated from marine algae, was reported to be a PDK3 inhibitor possibly interacting with the ATP binding site [35]. The ATP binding region of the PDKs, which is located at the end of the C-terminal domain, is a very conserved domain comprising four-strand mixed β-sheets and three α-helices [18]. IQ was predicted to bind two amino acid residues near the ATP binding site of PDK1 isoform. The results from biochemical ATP binding assay and Auto-ITC analysis using PDK1 supported the computational prediction. Thus, we suggested that IQ inhibits the PDK1 activity by interacting with the ATP binding pocket. To elucidate the isoform selectivity and precise mechanism of action underlying the inhibitory effect of IQ on PDKs, additional experiments, including an in vitro PDK activity assay, enzyme kinetic analysis, and crystallographic study, should be performed.

To validate the relative efficacy of IQ, the cytotoxic effect of IQ was compared with established PDK inhibitors including JX06 and AZD7545 that interact with the ATP binding pocket and lipoyl-binding pocket, respectively [36,37]. The results suggested that the anticancer efficiency of IQ is higher than that of JX06 and AZD7545 in A549 cells. The GI_50_ concentrations of hordenine, a small molecule PDK inhibitor that interferes with ATP binding, were reported to be 14.95 μM in A549 cells following 48-h treatment [35]. The derivatives of radicicol and M77976 showed potent cytotoxic activities on several cancer cell lines [34,38]. Radicicol, M77976, and their derivative compounds have a similar chemical structure, i.e., a resorcinol moiety [34]. Because this moiety is critical for their binding to the ATP binding site of heat shock protein 90 (Hsp90), radicicol and M77976 also inhibit the kinase activity of Hsp90 [34,39]. Although IQ does not have the resorcinol moiety, it has been reported to be an inhibitor of other enzymes, including *Mycobacterium tuberculosis* shikimate kinase, S-adenosylhomocysteine hydrolase, DNA polymerase β, and M-phase inducer phosphatase 2 [25,40,41]. It is well known that some natural products exert activities on multiple targets. Thus, defined novel approaches, such as synthetic modification, probe conjugation, and advanced-target elucidation, are needed to elucidate the global mechanism of action of natural products [42]. Therefore, we are trying to modify the chemical structure of IQ to develop a more potent anticancer drug and a specific PDK inhibitor. In addition, extensive preclinical experiments, including those assessing the pharmacodynamics, safety, and anticancer efficacy of IQ in animal models, should be conducted in future studies.

## 4. Materials and Methods

### 4.1. Materials

The antibody against phosphor-PDHA (#ab177461) was purchased from Abcam (Cambridge, UK). The antibodies against PDHA (#sc-377092), the PDC-E2 subunit (#sc-271352), and glyceraldehyde 3-phosphate dehydrogenase (GAPDH; #sc-32233) were purchased from Santa Cruz Biotechnology (Dallas, TX, USA). Antibodies against caspase 3 (#9665s) and 9 (#9508s) and poly ADP-ribose polymerase (PARP) (#9542s) were supplied by Cell Signaling Technology (Danvers, MA, USA). The antibody against PDK1 (#ADI-KAP-PK112) was purchased from Enzo Life Sciences (Farmingdale, NY, USA). The antibody against PDK3 (#32581) was purchased from NovusBio (Littleton, CO, USA). The antibodies against PDK2 (#41330) and 4 (#38562) were purchased from Signalway Antibody (College Park, MD, USA).

### 4.2. Purification and Validation of IQ 

The marine sponge (*S. cerebriformis*, 50 kg) was extracted with EtOH at room temperature for 7 days. The EtOH extract was filtered using a filter paper (Whatman No. 2) and then evaporated under reduced pressure. The dried extract (SCE, 3569 g) was separated via silica gel column chromatography using a step-gradient solvent of hexane and acetone (10:1–0:1) to obtain 10 fractions (SCE 1–10). Fraction SCE 5 (39.3 g) was further fractionated via silica gel column chromatography using a gradient system of hexane and acetone to obtain nine subfractions (SCE 5-1~SCE 5-9). Subfraction SCE 5-7 was subjected to C_18_ MPLC under isocratic solvent conditions (MeOH:H_2_O = 85:15) to obtain five fractions (SCE 5-7-1~SCE 5-7-5). Subfraction SCE 5-7-2 was subjected to reversed-phase HPLC using a MeOH-H_2_O (78:22) isocratic system to obtain IQ (110 mg, purity: 98.9%). The structure of IQ was determined via comparison of the various spectroscopic data with those in the literature [15]. IQ (**1**): Yellow powder; ^1^H NMR (CDCl_3_, 500 MHz) *δ* 2.08, 1.42 (each 1H, m, H-1), 1.84, 1.16 (each 1H, m, H-2), 2.29, 2.05 (each 1H, ddd, *J* = 13.7, 8.6, 5.4 Hz, H-3), 1.49, 1.32 (each 1H, m, H-6), 1.37 (2H, m, H-7), 1.14 (1H, m, H-8), 0.74 (1H, d, *J* = 12.0 Hz, H-10), 4.43, 4.41 (each 1H, s, H-11), 1.02 (3H, s, H-12), 0.96 (3H, d, *J* = 6.4, H-13), 0.82 (3H, s, H-14), 2.51, 2.45 (each 1H, d, *J* = 13.7, H-15), 5.83 (1H, s, H-19), 3.84 (3H, s, OCH_3_); ^13^C-NMR (CDCl_3_, 125 MHz) *δ* 23.3 (C-1), 28.1 (C-2), 33.1 (C-3), 160.7 (C-4), 40.6 (C-5), 36.8 (C-6), 28.8 (C-7), 38.3 (C-8), 43.5 (C-9), 50.3 (C-10), 102.7 (C-11), 20.7 (C-12), 18.0 (C-13), 17.5 (C-14), 32.5 (C-15), 117.5 (C-16), 153.5 (C-17), 182.5 (C-18), 102.2 (C-19), 161.9 (C-20), 182.2 (C-21), 57.0 (OCH_3_).

### 4.3. Cell Culture

The human lung cancer A549 cells, colorectal cancer DLD-1 cells, colon cancer RKO cells, embryonic kidney HEK293T cells, normal skin Detroit-551 cells, and murine Lewis lung carcinoma LLC cells were purchased from the American Type Culture Collection (Manassas, VA, USA). The DLD-1 cells, RKO cells, Detroit-551 cells, LLC cells, and HEK293T cells were maintained in Dulbecco’s modified Eagle’s medium (Welgene, Gyeongsan, Korea). The A549 cells were cultured in RPMI 1640 medium (HyClone™; GE Healthcare Life Sciences, Logan, UT, USA). All culture media were supplemented with 10% heat-inactivated fetal bovine serum (Welgene) and antibiotics (100 U/mL penicillin and 100 μg/mL streptomycin; Thermo Fisher Scientific, Rockford, IL, USA). The cells were maintained in a humidified 5% CO_2_ incubator at 37 °C.

### 4.4. Cell Viability Assay

The cytotoxic effect of IQ was examined using the 3-(4,5-dimethylthiazol-2-yl)-2,5-diphenyltetrazolium bromide (MTT) assay. Briefly, the A549 cells (10^4^ cells per well) were seeded in a 96-well plate. On the following day, the cells were treated with IQ. After 24 h, the MTT solution (2 mg/mL) was added to the cells. The absorbance of the developed formazan crystals was measured using a Spectramax M2 microplate reader (Molecular Devices, Sunnyvale, CA, USA) at 540 nm.

### 4.5. Western Blotting Assay

The Western blotting assay was performed as described previously [12]. Briefly, similar amounts (20 μg) of whole-cell lysates were separated by sodium dodecyl sulfate–polyacrylamide gel electrophoresis. The separated proteins were then transferred onto a nitrocellulose membrane (Hybond ECL; GE Healthcare, Menlo Park, CA, USA), which was subsequently incubated with antibodies. The target proteins were detected using ECL Plus reagent and then, analyzed using ImageQuant LAS 4000 (GE Healthcare).

### 4.6. Lactate Production Assay

Secretory lactate levels in the cell culture medium were measured using a lactate fluorometric assay kit (Biovision, Milpitas, CA, USA) [43] and analyzed using the Spectramax M2 microplate reader at 570 nm.

### 4.7. O_2_ Consumption Assay

The O_2_ consumption rate was measured using an Oxygen Consumption Rate Assay Kit (Cayman Chemical, Ann Arbor, MI, USA) according to the manufacturer’s protocol. The oxygen probe was analyzed using the Spectramax M2 microplate reader at wavelengths of 380/650 nm.

### 4.8. CRISPR-Mediated Genome Editing of PDHA1

The target sequences for CRISPR interference were designed on the CRISPR Design Tool (https://zlab.bio/guide-design-resources) provided by Zhang Lab (Cambridge, MA, USA). The target sequence for PDHA1 is GATGCAGACTGTACGCCGAA (exon 4). A complementary oligonucleotide with BpiI restriction sites for guide RNAs (gRNAs) was synthesized by Macrogene (Seoul, Korea) and cloned into the pX459 CRISPR/Cas9-puro vector (Addgene, Cambridge, MA, USA) [44].

### 4.9. Transient Transfection

The A549 cells were transfected with pX459-sgRNA using Lipofectamine2000 (Invitrogen, Karlsruhe, Germany) according to the manufacturer’s instructions. Starting on the day after transfection, these cells were treated with 5 μg/mL puromycin (CAS58-58-2; Santa Cruz Biotechnology) for 2 days. The surviving cells were reseeded at 1 cell per well in a 96-well plate using an Automated High-speed Cytometry Sorter System (BD FACS Aria III; BD Biosciences). To select the PDHA1-knockout colonies, the expression of PDHA1 was confirmed by Western blotting with anti-PDHA1. In selected colonies, the genome sequence of the edited locus was confirmed by sequence analyses at Bioneer Corporation (Daejeon, Korea).

### 4.10. T7 Endonuclease I Assay

Genomic DNA was purified from the target cells. PCR amplifications were carried out using the PCR system described above. The extension reaction was initiated by pre-heating the reaction mixture to 95 °C for 10 min; 30 cycles of 95 °C for 45 s, 60 °C for 1 min; and 72 °C for 45 s using the appropriate primers. The PCR products were evaluated via agarose gel electrophoresis and purified using the QIAquick™ Gel extraction kit (Qiagen, Hilden, Germany). Next, the PCR products were annealed in a thermocycler and incubated with 1 μL of T7 Endonuclease at 37 °C for 15 min. Nuclease-specific cleavage products were then determined by agarose gel electrophoresis.

### 4.11. ATP binding Assay

The cloned PDK1 was expressed in *Escherichia coli* BL21(DE3). The overexpressed protein was purified using affinity chromatography and gel filtration chromatography. The purified His-tagged PDK1 was bound to Ni-NTA beads in binding buffer (20 mM HEPES-K^+^ (pH 7.2) and 0.05% BSA). The Ni-NTA beads with bound His-tagged PDK1 were washed with binding buffer followed by incubation with (α-^32^P)ATP, ATP, and IQ at 37°C. The beads were then washed with wash buffer. The bead-bound PDK1 protein was obtained using 250 mM imidazole, and its radioactivity was detected by liquid scintillation counting.

### 4.12. PDH Complex and PDK Interaction Assay

The binding assay was performed as described previously [2]. Briefly, the HEK293T cells were transfected with the GST-PDK1 construct and then lysed. The lysate was subsequently incubated with Glutathione Sepharose 4B beads and washed. HEK293T whole-cell lysates containing the PDH complex with or without 50 μM IQ were added to the beads. The samples containing or not containing 50 μM IQ were subjected to Western blotting assay.

### 4.13. Auto-ITC

The dissociation constant and stoichiometry between the His-tagged PDK1 and IQ were determined via auto-isothermal titration calorimetry. The proteins were dialyzed in buffer (50 mM Tris-HCl (pH 7.5) and 200 mM NaCl) at a concentration of 0.1 mM. IQ was solubilized in the same buffer at a concentration of 1.0 mM. Titrations consisting of 20 injections at a 200-s interval were performed at 25 °C while the syringe was stirred at 1000 rpm. The determined K and ΔH values were used to calculate ΔS from the standard thermodynamic equation. Auto-ITC experiments were conducted using a MicroCal AutoITC200 (GE Healthcare, Stockholm, Sweden), and the data were analyzed using the Origin 7.0 program (OriginLab Corporation, Northampton, MA, USA).

### 4.14. Structural Prediction of the PDK1 and IQ Interaction

The crystal structure of the PDK1 protein (ID: 2Q8F) and the structure of IQ (CID: 21727418) were used for structural prediction. The research design was used to computationally determine the potential activity and binding affinity of IQ to the PDK1 protein. The structural prediction of the PDK1 and IQ complex was performed in Pyrx program. The binding affinity of IQ to PDK1 was determined to be −6.9 kcal/mol.

### 4.15. Mitochondrial Reactive Oxygen Species (ROS) Assay

The A549 cells (5 × 10^5^ cells per well) were seeded in a 6-well plate. On the following day, the cells were treated with 50 μM IQ for 12 h. Then, the cells were incubated with 5 μM MitoSOX™ Red (Thermo Fisher Scientific) for 30 min and analyzed via FACS (BD FACS CANTO II, BD Biosciences, Sunnyvale, CA, USA) at excitation and emission wavelengths of 510 and 580 nm, respectively.

### 4.16. Mitochondrial Depolarization Assay

The A549 cells (5 × 10^5^ cells) were seeded in a 6-well plate. Then, the cells were treated with the indicated concentrations of IQ for 12 h and subsequently incubated with 50 nM tetramethylrhodamine methyl ester (TMRM) (Thermo Fisher Scientific) for 30 min. The fluorescence intensities of the cells were measured using a fluorescence microscope and FACS at excitation and emission wavelengths of 510 and 580 nm, respectively.

### 4.17. Annexin V and PI Staining

The A549 cells (5 × 10^5^ cells) were seeded in a 6-well plate and then treated with the indicated concentrations of IQ for 24 h. The cells were subsequently examined using an Annexin V-FITC Apoptosis Detection kit (Life Technologies, Carlsbad, CA, USA). The fluorescence intensities of the cells were analyzed using FACS.

### 4.18. Statistical Analysis

The values obtained from the cell viability experiments and the (α-^32^P) ATP-bound levels were determined as percentages of the control and expressed as the mean ± standard deviation (SD). The relative phosphor-PDHA levels, lactate levels, O_2_ consumption rate, mitochondrial ROS levels, TMRM fluorescence levels, and apoptotic population rate were calculated as fold changes of the control. For comparisons of two groups, the difference in the mean values was calculated by Student’s *t*-test. The significance of serial data was verified by a one-way analysis of variance with Tukey’s post hoc test. All experiments were independently performed at least thrice. The GI_50_ values of IQ were determined by the sigmoidal fitting of the cell viability data in the GraphPad Prism program (GraphPad Software, La Jolla, CA, USA).

## 5. Conclusions

In conclusion, to our knowledge, this is the first study reporting that IQ has a potent inhibitory effect on the activity of PDK1. Subsequently, IQ also induced the mitochondrial ROS-mediated apoptosis of human lung cancer A549 cells. Our findings provide evidence that IQ might be a candidate antimetabolic cancer drug. In addition, we suggest IQ as a lead compound for the development of a potent and specific PDK inhibitor.

## Figures and Tables

**Figure 1 ijms-21-06021-f001:**
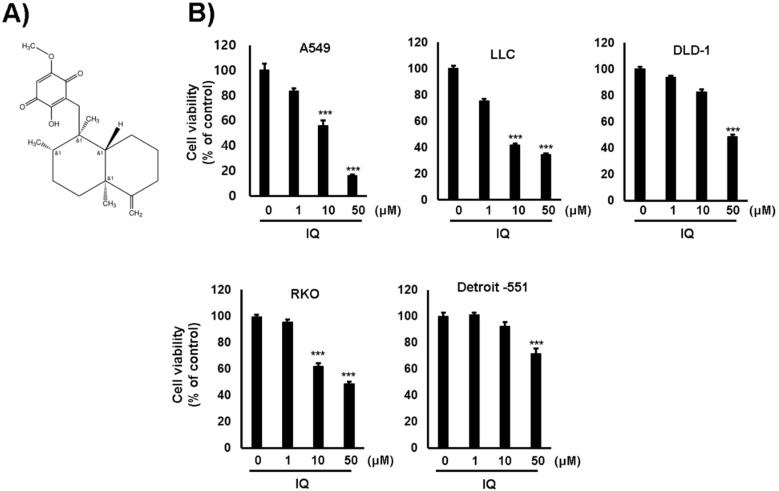
IQ reduced the cell viability of several cancer cell lines. (**A**) The chemical structure of IQ is shown. (**B**) The indicated cell lines were treated with IQ for 24 h. The results of the MTT assay are shown. Cell viability is shown as the mean ± SD. The cell viability assay was performed at least three times. Statistical analyses were performed using Student’s *t*-test. ***, *p* < 0.001 compared with the control.

**Figure 2 ijms-21-06021-f002:**
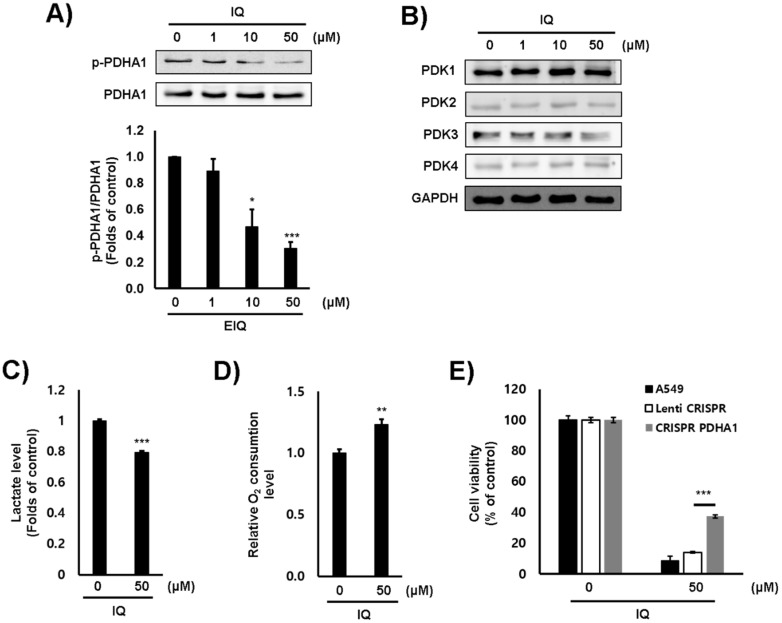
IQ reduced PDK activity and promoted OxPhos in A549 cells. The A549 cells were treated with IQ for 4 h. (**A**,**B**) The levels of phosphorylated PDHA1 (**A**) and PDK1–4 (**B**) were analyzed via Western blotting. ((**A**), **lower panel**) The bar graph from three independent experiments ((**A**), **upper panel**) is shown; PDHA and GAPDH were used as loading controls. (**C**) Lactate production after IQ treatment was measured using a lactate fluorometric assay kit. (**D**) The cells were treated with IQ, and the O_2_ consumption rate was measured using a commercially available Oxygen Consumption Rate Assay Kit. (**E**) The cells were treated with the indicated concentrations of IQ for 24 h. The viability of these cells was measured using the MTT assay. All experiments except for panel B were performed at least three times, and data are shown as the mean ± SD. Statistical analyses were performed using Student’s *t*-test. *, *p* < 0.01 **, *p* < 0.05 and ***, *p* < 0.001 compared with the control.

**Figure 3 ijms-21-06021-f003:**
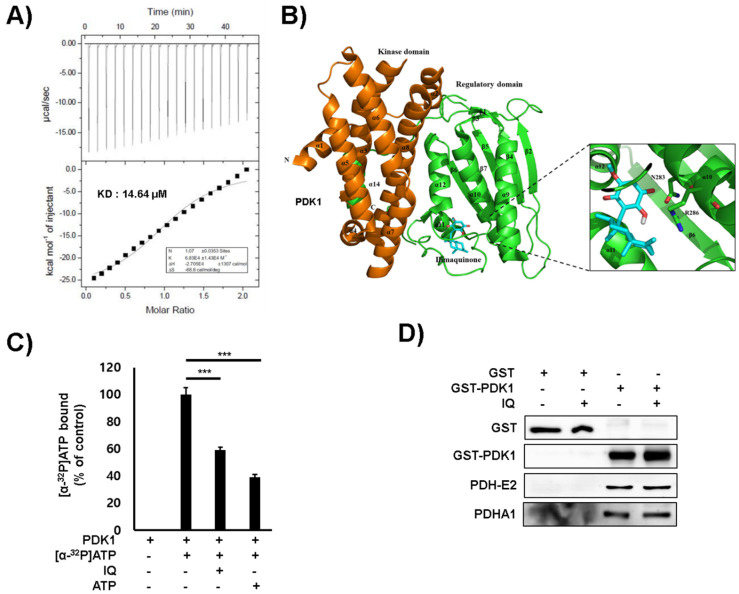
IQ interacted with the ATP binding pocket of PDK1. (**A**) The ITC analysis results of PDK1 and IQ are shown. IQ was titrated into PDK1 solutions, and the measured Kd values are shown. (**B**) IQ binds to the ATP binding pocket of PDK1. The modeled structure of PDK1 with IQ is shown as a ribbon diagram. The interaction residues between PDK1 and IQ are shown. (**C**) The ATP binding assay results of PDK1 and IQ are shown. The binding of ATP to PDK1 in the presence of IQ was determined using α^32^-labeled ATP. Non-bound α-^32^P-GTP was washed out, and the radioactivity was examined using a scintillation counter. Non-labeled ATP was used as a competitor. Data are presented as the mean ± SD. Statistical analyses were performed using Student’s *t*-test. *** *p* < 0.001 compared with the control under α^32^-labeled ATP (second column). (**D**) The PDC subunits and PDK binding were analyzed.

**Figure 4 ijms-21-06021-f004:**
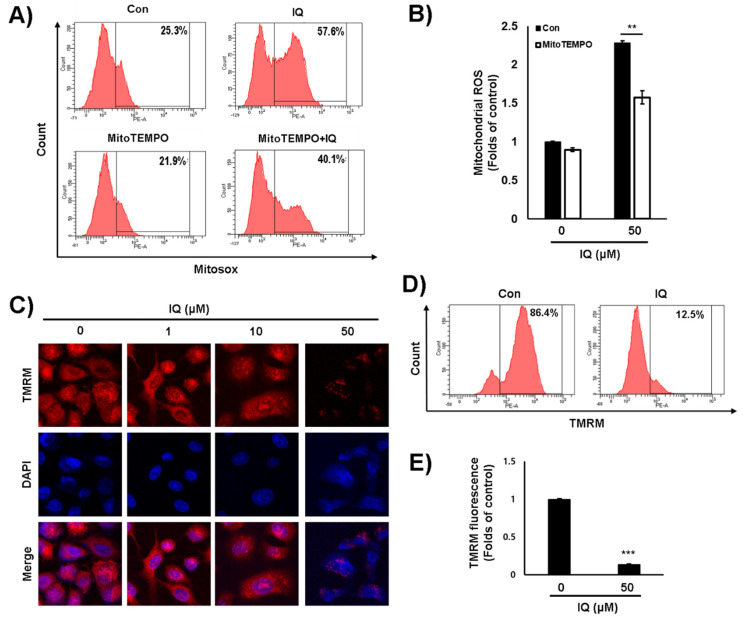
IQ increased mitochondrial ROS and mitochondrial damage in A549 cells. (**A**,**B**) The A549 cells were pretreated with MitoTEMPO for 1 h at 50 μM and treated with IQ for 12 h at 50 μM. (**A**) After the cells were stained with MitoSOX™ Red, the fluorescence was analyzed by FACS. (**B**) The MTT assay was performed. The cells were treated with IQ for 12 h and then, stained with TMRM. The fluorescence intensity was analyzed via confocal microscopy and FACS (**C**,**D**). (**E**) The bar graph from three independent experiments (**D**) is shown, and the values represent the mean ± SD. The experiments of panel B and E were performed at least three times, and data are shown as the mean ± SD. Statistical analyses were performed using Student’s *t*-test. **, *p* < 0.05 and ***, *p* < 0.001 compared with the control.

**Figure 5 ijms-21-06021-f005:**
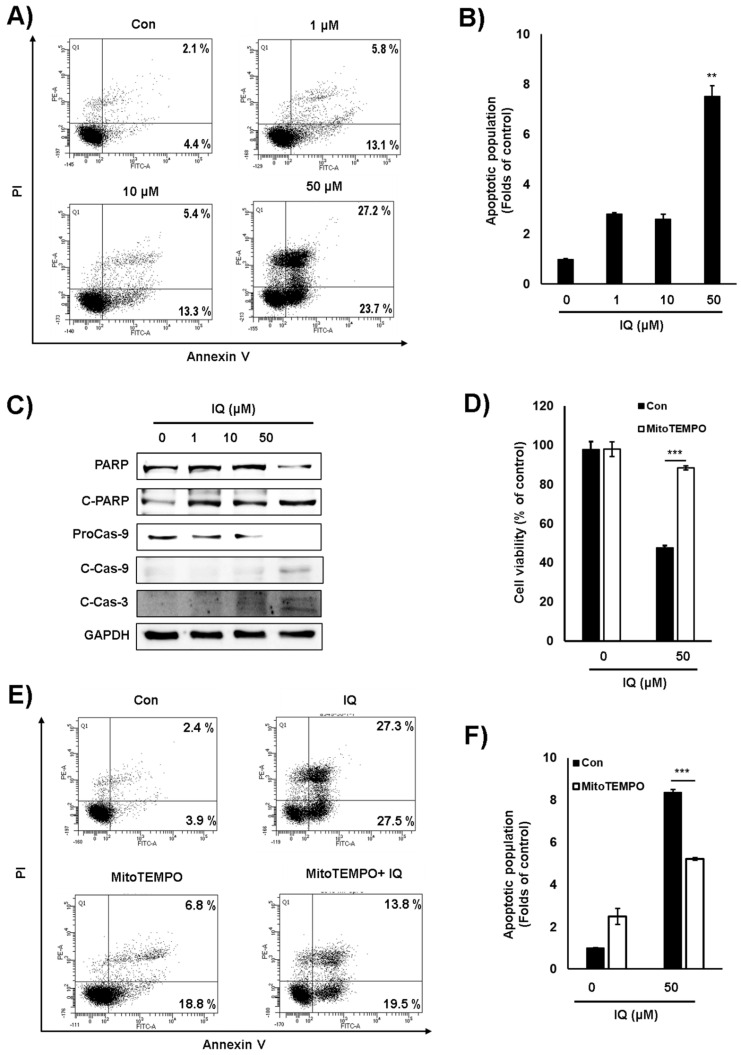
IQ induced apoptosis via mitochondrial ROS production in A549 cells. (**A**) The A549 cells were treated with IQ for 24 h. The cells were stained with annexin V-PI and analyzed via FACS analysis. (**B**) The bar graph is shown in panel A. (**C**) The expression of caspase 3, caspase 9, and PARP in the cells was analyzed via Western blotting. GAPDH was used as a loading control. (**D**–**F**) The cells were pretreated with MitoTEMPO for 1 h and then, treated with IQ for 18 h. (**D**) The cells were analyzed via MTT assay. (**E**) The cells were examined via annexin V-PI analysis. (**F**) The bar graph is shown in panel E. The experiments of panel B, D, and F were performed at least three times, and data are shown as the mean ± SD. Statistical analyses were performed using Student’s *t*-test. **, *p* < 0.05 and ***, *p* < 0.001 compared with the control.

**Table 1 ijms-21-06021-t001:** GI_50_ values of the effect of 24-h IQ treatment on cell viability.

Cell Line	GI_50_ (μM)
A549	10.5 ± 3.71
LLC	8.612 ± 4.208
DLD-1	50.16 ± 9.78
RKO	37.3 ± 27.19
Detroit 551	>100

The experiments were performed at least three times, and data are shown as the mean ± SD.

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
