# Peer review of "Ilimaquinone Induces the Apoptotic Cell Death of Cancer Cells by Reducing Pyruvate Dehydrogenase Kinase 1 Activity"

_ijms, 2020, doi:10.3390/ijms21176021_

Round 1
Reviewer 1 Report
The manuscript by Kwak and co-workers reports the finding that ilimaquinone (IQ) is an inhibitor of pyruvate dehydrogenase kinase 1 activity. They provide evidence that IQ inhibits PDH phosphorylation, IQ binds to PDK1, and IQ inhibits cell viability, increases O2 consumption and decreases lactate production. They concluded that IQ might be a novel candidate for anticancer therapeutics by inhibiting PDK1
There are several major issues with this manuscript.
1) The introduction does not provide sufficient background on PDK1 in its relationship with cancer, and on previous efforts in developing inhibitors against PDK1.
2) The results section is too brief, and does not provide sufficient information about the experiments or the results. It only stated the conclusions from the experiments.
3) Table 1 reports the 50% growth inhibition (GI50), but did not define growth inhibition. It is a 24 h endpoint assay, what was determined appeared to be viability. How was growth inhibition calculated? Further, the GI50's do not have any statistical analysis. How many times were the experiments done?
4) The best inhibition was for A549 cells with a reported GI50 of 24 microM. This level of potency is far weaker than that of available PDK1 inhibitors, such as AZD7545 and others. That make this inhibitor potentially much less useful.
5) The manuscript claims that IQ affects the cell viability through inhibiting PDK1. What is the evidence to exclude the other isozymes?
6) In Fig 2E, with PDHA1 knockout, IQ still significantly inhibits cell viability, although less potently than the wt. This needs to be explained. How did PDHA1 knockout affect cell viability? This information is also relevant to this result.
7) How is this finding related to other reports that IQ affects cell viability through other mechanisms in refs 7-9? If this inhibitor can affect many unrelated cellular processes, it also makes it less useful as a cancer drug because of potential toxicity issues.
8) How does this inhibitor compare to other PDK1 inhibitors. Including one or more established inhibitors in some experiments will make the conclusion more convincing.
A minor error: Fig 2C and 2D are switched.
Author Response
Response to Reviewer #1:
Comment:
The manuscript by Kwak and co-workers reports the finding that ilimaquinone (IQ) is an inhibitor of pyruvate dehydrogenase kinase 1 activity. They provide evidence that IQ inhibits PDH phosphorylation, IQ binds to PDK1, and IQ inhibits cell viability, increases O2 consumption and decreases lactate production. They concluded that IQ might be a novel candidate for anticancer therapeutics by inhibiting PDK1
There are several major issues with this manuscript.
Answer: We are very thanks to Reviewer’s valuable and constructive comments. The manuscript was modified according to the comments. We think that our manuscript was improved by the revision.
Comment: 1) The introduction does not provide sufficient background on PDK1 in its relationship with cancer, and on previous efforts in developing inhibitors against PDK1.
Answer: According to Reviewer’s comment, we revised the 1st paragraph of introduction section and added the 2nd paragraph as follows.
It is known that, under oxygen-abundant conditions, glycolysis rather than oxidative phosphorylation (OxPhos) is normally preferred as the energy source in cancer cells; this phenomenon is termed the Warburg effect [1]. Pyruvate dehydrogenase kinases (PDKs) are the primary metabolic enzymes involved in the Warburg effect. Four different isoforms of PDK have been found in mammals, and these enzymes regulate the activity of the pyruvate dehydrogenase complex (PDC) by phosphorylating its E1 alpha subunit (PDHA1) [2]. The PDC is divided into mainly three parts: pyruvate dehydrogenase E1 subunit, dihydrolipoamide acetyltransferase (the E2 subunit), and lipoamide dehydrogenase (the E3 subunit); PDKs bind to the E2 subunit and phosphorylate PDHA1 [3]. The PDC converts pyruvate into acetyl-CoA, which enters the tricarboxylic acid cycle; thus, the activity of the PDC is very important for OxPhos.
It is well known that, in many types of cancers, PDKs, especially PDK1, are overexpressed and/or its activity is increased by post-translational modifications [4-6]. In addition, PDK1 is highly expressed in non-small cell lung cancer (NSCLC), and its overexpression is significantly correlated with poor prognosis in human NSCLC tissues [7]. Furthermore, the levels of PDK1 and phosphor-PDHA1 are regarded as prominent negative prognostic factors in NSCLC [8]. Pyruvate could be converted to lactate by lactate dehydrogenase A (LDHA) when the activity of PDH complex was reduced by PDK1 activation [9]. The reprogramming of cancer metabolism through the enhancement of PDC activity via the inhibition of PDK1 increases the anticancer effects on several cancer cell lines, including NSCLC [10, 11]. Thus, our group has continued the efforts to discover novel PDK1 inhibitors from natural compounds as candidates of anticancer drugs [2, 12, 13].
Comment: 2) The results section is too brief, and does not provide sufficient information about the experiments or the results. It only stated the conclusions from the experiments.
Answer: As Reviewer’s suggested, we added the description in results section as follows.
In results 2.1. section,
Firstly, the cytotoxicity of IQ on several lung and colorectal cancer cell lines was examined. The viability of each cell line was significantly decreased by IQ treatment (Figure 1). The 50% growth inhibition (GI50) values of IQ were shown in Table 1. Because the GI50 value of IQ in the A549 lung cancer cells was the lowest, further investigation was performed using this cell line. In addition, the cell viability was markedly reduced by IQ treatment when compared with that by JX06 and AZD8545, well-known PDK inhibitors (Supplementary Figure S1). Next, the inhibitory effect of IQ on the activity of PDK1 was estimated via Western blotting using a phosphor-PDHA1 antibody. The phosphorylation of PDHA1 was markedly reduced by IQ treatment up to 50 μM for 4 h (Figure 2A). At the condition, the survival of A549 cells did not show a significant reduction (data not shown). To rule out whether the reduction of PDHA1 phosphorylation was due to PDKs expression, the protein levels of PDKs were examined. IQ did not affect the expression of PDK1–4 in the conditions (Figure 2B). These data suggested that IQ decreased PDHA1 phosphorylation through suppressing the activity of PDKs but not affecting their expression. Furthermore, because PDHA1 activation promotes the OxPhos, we examined the O2 consumption and secreted lactate levels. IQ decreased lactate production and increased O2 consumption in the A549 cells (Figure 2C and D). To ensure whether the inhibition of PDK activity is directly correlated with the cytotoxic effect of IQ, we constructed PDHA1-knockout A549 cells using a clustered regularly interspaced short palindromic repeats (CRISPR) system (Supplementary Figure S2). The proliferation of PDHA1-knockout cells did not show a significant difference comparing with parent A549 cells (data not shown). However, the cytotoxicity of IQ was significantly recovered in the PDHA1-knockout cells (Figure 2E). From these results, we supposed that IQ could increase PDH activity and OxPhos, and subsequently reduce the cell viability via suppressing PDKs activity.
In results 2.3. section,
It was reported that the inhibition of PDK1 activity induces the production of mitochondrial ROS in cancer cells [19, 20]. As in previous studies, the mitochondrial ROS was significantly increased in the presence of IQ (Figure 4A and 4B). Moreover, MitoTEMPO, a mitochondrial ROS inhibitor, significantly decreased the IQ-induced increase in mitochondrial ROS production (Figure 4A and B). When mitochondrial ROS increases, the membrane potential of mitochondria consequently decreased [2]. Thus, we examined the effect of IQ on the membrane potential of mitochondria using TMRM staining. The mitochondrial membrane potential was also markedly depolarized by IQ treatment using confocal microscope imaging (Figure 4C) and FACS analysis (Figure 4D, and E). These results suggested that IQ significantly induces the mitochondrial damage by inhibiting the PDK1 activity.
To confirm whether the type of cell death induced by IQ was apoptosis, we examined the annexin V-propidium iodide (PI) double staining. As shown Figure 5A, and B, IQ significantly induced the population of annexin V-PI positive cells. Furthermore, as determined by Western blot analysis, IQ significantly induced cleavage of caspase-3, -9, and PARP in A549 cells (Figure 5C). These data suggested that IQ induce apoptosis in A549 cells. To confirm the mitochondrial ROS-dependent apoptosis, we examined the cell viability and annexin V-PI staining in the presence of MitoTEMPO and IQ. The pretreatment with MitoTEMPO reduced the cytotoxic effect of IQ, and the population of apoptotic cells was clearly rescued (Figure 5D-F). These results suggested that IQ induce the mitochondrial ROS-dependent apoptosis in A549 cells.
Comment: 3) Table 1 reports the 50% growth inhibition (GI50), but did not define growth inhibition. It is a 24 h endpoint assay, what was determined appeared to be viability. How was growth inhibition calculated? Further, the GI50's do not have any statistical analysis. How many times were the experiments done?
Answer: We appreciate Reviewer’s careful indication. We found out that we made a mistake calculating the GI50 values. We recalculated the GI50 values in Table 1 and added the description in material and methods section as follows.
In materials and methods section,
The GI50 values of IQ were determined by the sigmoidal fitting of the cell viability data in the GraphPad Prism program (GraphPad Software, La Jolla, CA, USA).
In Table 1,
Table 1. GI50 values of the effect of 24-h IQ treatment on cell viability.
|
Cell line |
GI50 (μM) |
|
A549 |
10.5 ± 3.71 |
|
LLC |
8.612 ± 4.208 |
|
DLD-1 |
50.16 ± 9.78 |
|
RKO |
37.3 ± 27.19 |
|
Detroit 551 |
> 100 |
The experiments were performed at least three times, and data shown as the mean ± SD.
Comment: 4) The best inhibition was for A549 cells with a reported GI50 of 24 microM. This level of potency is far weaker than that of available PDK1 inhibitors, such as AZD7545 and others. That make this inhibitor potentially much less useful.
Answer: According to Reviewer’s comment 3, we recalculated the GI50 value. The value on A549 cells for 24 h treatment of IQ is 10.5 μM. In addition, we performed additional experiments using JX06 and AZD7545 and the results were shown in Supplementary Figure S1. JX06, a PDK1 inhibitor interacting with ATP binding pocket, was reported that growth inhibition IC50 values were lower than 0.5 μM for 72 h treatment on eight different cancer cell lines [1]. In this study, when JX06 was treated in A549 cells for 24 h (the same conditions of IQ treatment), JX06 did not show superior effect on growth inhibition. In case of AZD7545, a well-known PDK4-specific inhibitor interfering the lipoamide-binding pocket. As shown Supplementary Figure S1, AZD7545 did not show significant cytotoxic effect on A549 cells.
Supplementary Figure S1. The cell viabilities of JX06, and AZD7545 in A549 cells. The cells were treated with JX06, and AZD7545 for 24 h. All experiments were performed at least three times, and data shown as the mean ± SD. Statistical analyses were performed using Student’s t-test. ***, p < 0.001 compared with the control.
Similar to our data, Zhang et al [2] showed that high concentration of AZD7545 (400 μM) did not show an anticancer effect in RKO colorectal cancer cells. In addition, other Zhang et al [3] also reported that AZD7545 has no anticancer effect on A549 and other cells. Thus, we carefully suggested that GI50 value of IQ in A549 cells is relevant concentration.
We added the description in discussion section as follows.
To validate the relative efficacy of IQ, the cytotoxic effect of IQ was compared with established PDK inhibitors including JX06 and AZD7545 that interact with the ATP binding pocket and lipoyl-binding pocket, respectably. The results suggested that the anticancer efficiency of IQ is higher than that of JX06 and AZD7545 in A549 cells.
References
- Sun, W.; Xie, Z.; Liu, Y.; Zhao, D.; Wu, Z.; Zhang, D.; Lv, H.; Tang, S.; Jin, N.; Jiang, H., JX06 selectively inhibits pyruvate dehydrogenase kinase PDK1 by a covalent cysteine modification. Cancer research 2015, 75, (22), 4923-4936.
- Zhang, S.-L.; Zhang, W.; Yang, Z.; Hu, X.; Tam, K. Y., Synthesis and biological evaluation of (R)-3, 3, 3-trifluoro-2-hydroxy-2-methylpropionamides as pyruvate dehydrogenase kinase 1 (PDK1) inhibitors to reduce the growth of cancer cells. European Journal of Pharmaceutical Sciences 2017, 110, 87-92.
- Zhang, W.; Zhang, S. L.; Hu, X. H.; Tam, K. Y., Targeting Tumor Metabolism for Cancer Treatment: Is Pyruvate Dehydrogenase Kinases (PDKs) a Viable Anticancer Target? Int J Biol Sci 2015, 11, (12), 1390-1400.
Comment: 5) The manuscript claims that IQ affects the cell viability through inhibiting PDK1. What is the evidence to exclude the other isozymes?
Answer: We thanks for Reviewer’s thoughtful suggestion. We did not show the experimental results for the other isozymes in this study. However, based on structural prediction analysis shown in Figure 3 and Supplementary Figure S3, IQ could bind to ATP binding pocket of PDK1. It is well known that PDK1 is highly expressed in several cancers including non-small cell lung cancer and its expression is significantly correlated with poor prognosis in human non-small cell lung cancer tissues [4]. PDK2 and PDK3 were predicted to have only one interacting amino acid residue with IQ. In case of PDK4, it has three predictive ATP binding sites. However, PDK4 is frequently down-regulated in human cancer, especially in lung cancer [5]. Thus, in this study, we focused to PDK1 inhibition. From the computational prediction and biochemical analysis, we carefully assumed that PDK1 is the main target of IQ in human lung cancer A549 cells. And we modified the description in discussion section as follows.
In discussion section,
IQ was predicted to bind two amino acid residues near the ATP-binding site of PDK1 isoform. The results from biochemical ATP-binding assay and Auto-ITC analysis using PDK1 supported the computational prediction. Thus, we suggested that IQ inhibits the PDK1 activity by interacting with the ATP binding pocket. To elucidate the isoform selectivity and precise mechanism of action underlying the inhibitory effect of IQ on PDKs, additional experiments, including an in vitro PDK activity assay, enzyme kinetic analysis, and crystallographic study, should be performed.
References
- Liu, T.; Yin, H., PDK1 promotes tumor cell proliferation and migration by enhancing the Warburg effect in non-small cell lung cancer. Oncology reports 2017, 37, (1), 193-200.
- Sun, Y.; Daemen, A.; Hatzivassiliou, G.; Arnott, D.; Wilson, C.; Zhuang, G.; Gao, M.; Liu, P.; Boudreau, A.; Johnson, L., Metabolic and transcriptional profiling reveals pyruvate dehydrogenase kinase 4 as a mediator of epithelial-mesenchymal transition and drug resistance in tumor cells. Cancer & metabolism 2014, 2, (1), 1-14.
Comment: 6) In Fig 2E, with PDHA1 knockout, IQ still significantly inhibits cell viability, although less potently than the wt. This needs to be explained. How did PDHA1 knockout affect cell viability? This information is also relevant to this result.
Answer: We appreciate Reviewer’s careful indication and fully agree with Reviewer’s comment. In this experiment, our aim is to confirm whether cytotoxic effect of IQ is closely related with PDK inhibition. The pyruvate dehydrogenase (PDH) functions as a gatekeeper in pyruvate metabolism for converting to Acetyl-CoA. In many cancer cells, the PDH was frequently inactivated by phosphorylation of its E1alpha subunit (PDHA1) by kinase activity of PDKs, especially PDK1. To study the metabolic role of PDH complex and its kinase PDKs, the PDHA1 expression was genetically abolished in several previous studies [6-8]. In this study, we also adopted the strategy to reveal the correlation between PDK inhibition and cytotoxic effect in IQ-treated cancer cells. As shown in Figure 2E, the cytotoxic effect of IQ was significantly recovered in PDHA1-knockout cells. However, as indicated by Reviewer, although in PDHA1-knockout cells, IQ could inhibit the cell viability. The cell viability was recovered about 40% compared with that of wild type cells. The incomplete recovery of cell viability might due to existence of another targets of PDK1, including EGFR, HGF-MET, and PRAL, as well as PDHA1 [9-12], or to off targets effect of IQ. To elucidate the precise mechanism, it is required to perform further experiments. However, in this study, we carefully suggest that the cytotoxic effect of IQ was related with inhibition of PDK1 activity, at last partially.
We modified the description in discussion section as follows.
In discussion section,
To confirm whether the anticancer effect of IQ was related to inhibition of PDK1 activity, we used PDHA1-knockout A549 cells. In the PDHA1-knockout cells, cell viability was partially rescued by about 40% compared with IQ-treated wild type A549 cells. The results might due to off-targets of IQ or another targets of PDK1, including EGFR, HGF-MET, and PARL. Thus, we suggest that IQ significantly induced mitochondrial ROS production by inhibiting PDK1 activity and thereby induced apoptotic cell death in the A549 lung cancer cells.
References
- Li, Y.; Li, X.; Li, X.; Zhong, Y.; Ji, Y.; Yu, D.; Zhang, M.; Wen, J.-G.; Zhang, H.; Goscinski, M. A., PDHA1 gene knockout in prostate cancer cells results in metabolic reprogramming towards greater glutamine dependence. Oncotarget 2016, 7, (33), 53837.
- Jackson, L. E.; Kulkarni, S.; Wang, H. B.; Lu, J.; Dolezal, J. M.; Bharathi, S. S.; Ranganathan, S.; Patel, M. S.; Deshpande, R.; Alencastro, F.; Wendell, S. G.; Goetzman, E. S.; Duncan, A. W.; Prochownik, E. V., Genetic Dissociation of Glycolysis and the TCA Cycle Affects Neither Normal nor Neoplastic Proliferation. Cancer research 2017, 77, (21), 5795-5807.
- Li, Y. Q.; Li, X. R.; Li, X. L.; Zhong, Y. L.; Ji, Y. S.; Yu, D. D.; Zhang, M. Z.; Wen, J. G.; Zhang, H. Q.; Goscinski, M. A.; Nesland, J. M.; Suo, Z. H., PDHA1 gene knockout in prostate cancer cells results in metabolic reprogramming towards greater glutamine dependence. Oncotarget 2016, 7, (33), 53837-53852.
- Deng, X.; Wang, Q.; Cheng, M.; Chen, Y.; Yan, X.; Guo, R.; Sun, L.; Li, Y.; Liu, Y., Pyruvate dehydrogenase kinase 1 interferes with glucose metabolism reprogramming and mitochondrial quality control to aggravate stress damage in cancer. J Cancer 2020, 11, (4), 962-973.
- Huang, X.; Gan, G.; Wang, X.; Xu, T.; Xie, W., The HGF-MET axis coordinates liver cancer metabolism and autophagy for chemotherapeutic resistance. Autophagy 2019, 15, (7), 1258-1279.
- Zhang, M.; Cong, Q.; Zhang, X. Y.; Zhang, M. X.; Lu, Y. Y.; Xu, C. J., Pyruvate dehydrogenase kinase 1 contributes to cisplatin resistance of ovarian cancer through EGFR activation. Journal of Cellular Physiology 2019, 234, (5), 6361-6370.
- Shi, G.; McQuibban, G. A., The Mitochondrial Rhomboid Protease PARL Is Regulated by PDK2 to Integrate Mitochondrial Quality Control and Metabolism. Cell Rep 2017, 18, (6), 1458-1472.
Comment: 7) How is this finding related to other reports that IQ affects cell viability through other mechanisms in refs 7-9? If this inhibitor can affect many unrelated cellular processes, it also makes it less useful as a cancer drug because of potential toxicity issues.
Answer: We thanks for Reviewer’s constructive suggestion. We fully agree with Reviewer’s comment. We carefully discussed, the ER stress, autophagy, and p53-mediated cell death were previously reported as the molecular mechanism underlying IQ-induced cytotoxic effect. In addition, several studies demonstrated that relation between PDK activity and these cell death mechanisms including ER stress, autophagy, and p53. The further studies should be performed to reveal their relationships. In addition, we also think that IQ should be improved to develop a novel candidate for anticancer drug owing to limits of this study and general off target possibility of natural products. However, as mentioned in the last paragraph of Discussion section, we regarded IQ as a novel backbone for further chemical modification, because it did not share a chemical moiety with previous ATP-binding inhibitor on PDK1.
Thus, we added the description in discussion section as follows.
These studies suggest that complex molecular mechanisms might be related in IQ-induced cell death, such as PDK1 activity, ER stress, and p53 activity. To elucidate the precise mechanism underlying the anticancer mechanism of IQ with regard to ER stress, autophagy, and p53 regulation, extensive experiments are required.
Comment: 8) How does this inhibitor compare to other PDK1 inhibitors. Including one or more established inhibitors in some experiments will make the conclusion more convincing.
Answer: As kindly suggested by Reviewer, we examined the cell viability assay using established PDK1 inhibitors, such as AZD7545 and JX06. The results were shown in Supplementary Figure S1, as follow.
Supplementary Figure S1. The cell viabilities of JX06, and AZD7545 in A549 cells. The cells were treated with JX06, and AZD7545 for 24 h. The data are shown as the mean ± SD. ***, p < 0.001 compared with the IQ treated groups.
The results indicated that IQ has a superior cytotoxic effect in A549 cells compared with the established PDK1 inhibitor, JX06. AZD7545 did not significantly reduced the cell viability, as like previously reported.
Comment: A minor error: Fig 2C and 2D are switched.
Answer: As kindly pointed out by Reviewer, we modified the error as follow.
In Figure 2,
Figure 2. IQ reduced PDK activity and promoted OxPhos in A549 cells. The A549 cells were treated with IQ for 4 h. (A, B) The levels of phosphorylated PDHA1 (A) and PDK1–4 (B) were analyzed via western blotting. (A, lower panel) The bar graph from three independent experiments (A, upper panel) is shown, PDHA and GAPDH were used as loading controls. (C) Lactate production after IQ treatment was measured using a lactate fluorometric assay kit. (D) The cells were treated with IQ, and the O2 consumption rate was measured using a commercially available Oxygen Consumption Rate Assay Kit. (E) The cells were treated with the indicated concentrations of IQ for 24 h. The viability of these cells was measured using the MTT assay. All experiments except for penal B were performed at least three times, and data shown as the mean ± SD. Statistical analyses were performed using Student’s t-test. *, P < 0.01 **,p < 0.05 and ***, p < 0.001 compared with the control.
Reviewer 2 Report
The manuscript entitled ‘Ilimaquinone induces the apoptotic cell death of cancer cells by reducing pyruvate dehydrogenase kinase 1 activity’ by Kwak et al. describes experiments aimed at elucidation of the mechanism of anti-tumor cell effects of marine sponge derived natural product ilimaquinone. The manuscript is well written and follows a logical progression in experiments to achieve the goal of the study. The authors employ a variety of appropriate experiments with appropriate controls. While the studied biological activity of the titled compound as well as its binding affinity to the identified target are modest the authors do not over-interpret the results and suggest the need for additional experiments to fully elucidate mechanistic details. This manuscript should be of interest to the general drug design and cancer research communities.
This reviewer makes the following suggestions (corrections) to the authors.
Line 41. ‘PDHs’ should be ‘PDHAs’
Line 121. The word ‘considerable’ is not consistent with ‘slightly’ used earlier in the sentence
Figures 2, 4, and 5. ‘**’ and ‘***’ should both be defined in legend
Figure 5F. p-value in legend does not appear to be consistent with ‘***’ on graph, please check and correct
References. Reviewer found several entries where capitalization/punctuation was not consistent throughout listings. Please check carefully.
Author Response
Response to Reviewer #2:
Comments:
The manuscript entitled ‘Ilimaquinone induces the apoptotic cell death of cancer cells by reducing pyruvate dehydrogenase kinase 1 activity’ by Kwak et al. describes experiments aimed at elucidation of the mechanism of anti-tumor cell effects of marine sponge derived natural product ilimaquinone. The manuscript is well written and follows a logical progression in experiments to achieve the goal of the study. The authors employ a variety of appropriate experiments with appropriate controls. While the studied biological activity of the titled compound as well as its binding affinity to the identified target are modest the authors do not over-interpret the results and suggest the need for additional experiments to fully elucidate mechanistic details. This manuscript should be of interest to the general drug design and cancer research communities.
This reviewer makes the following suggestions (corrections) to the authors.
Answer: Thanks for your kind comment. We have revised the manuscript according to Reviewer’s comments. The comments are very valuable and constructive to improve the quality of our manuscript.
Comment: Line 41. ‘PDHs’ should be ‘PDHAs’
Answer: Thanks for your kind comment. As pointed out by Reviewer, we have a mistake in the description at line 39-41. Thus, we corrected the sentence as follow:
Four different isoforms of PDK have been found in mammals, and these enzymes regulate the activity of the pyruvate dehydrogenase complex (PDC) by phosphorylating its E1 alpha subunit (PDHA1)
In addition, we found the misspell of ‘PHDA1’ at line 67. Thus, we modified the word in results section as follows.
In the PDHA1-knockout cells, the cytotoxicity of IQ was significantly reduced (Figure 2E).
Comment: Line 121. The word ‘considerable’ is not consistent with ‘slightly’ used earlier in the sentence.
Answer: According to Reviewer’s comment, we modified the description in discustion section as follows.
“The GI50 values of IQ following 24-h treatment, as confirmed in this study, were slightly higher than those reported in the previous study, but the difference was slight.”
Comment: Figures 2, 4, and 5. ‘**’ and ‘***’ should both be defined in legend
Answer: According to Reviewer’s comment, we added the description in Figures and legends section as follows.
Figure 2. IQ reduced PDK activity and promoted OxPhos in A549 cells. The A549 cells were treated with IQ for 4 h. (A, B) The levels of phosphorylated PDHA1 (A) and PDK1–4 (B) were analyzed via western blotting. (A, lower panel) The bar graph from three independent experiments (A, upper panel) is shown, PDHA and GAPDH were used as loading controls. (C) Lactate production after IQ treatment was measured using a lactate fluorometric assay kit. (D) The cells were treated with IQ, and the O2 consumption rate was measured using a commercially available Oxygen Consumption Rate Assay Kit. (E) The cells were treated with the indicated concentrations of IQ for 24 h. The viability of these cells was measured using the MTT assay. All experiments except for penal B were performed at least three times, and data shown as the mean ± SD. Statistical analyses were performed using Student’s t-test. *, P < 0.01 **,p < 0.05 and ***, p < 0.001 compared with the control.
Figure 4. IQ increased mitochondrial ROS and mitochondrial damage in A549 cells. (A and B) The A549 cells were pretreated with MitoTEMPO for 1 h at 50 μM and treated with IQ for 12 h at 50 μM. (A) After the cells were stained with MitoSOX™ Red, the fluorescence was analyzed by FACS. (B) The MTT assay was performed. The cells were treated with IQ for 12 h and then stained with TMRM. The fluorescence intensity was analyzed via confocal microscopy and FACS (C and D). (E) The bar graph from three independent experiments (D) is shown, and the values represent the mean ± SD. The experiments of penal B, and E were performed at least three times, and data shown as the mean ± SD. Statistical analyses were performed using Student’s t-test. *, P < 0.01 **, p < 0.05 and ***, p < 0.001 compared with the control.
Figure 5. IQ induced apoptosis via mitochondrial ROS production in A549 cells. (A) The A549 cells were treated with IQ for 24 h. The cells were stained with annexin V-PI and analyzed via FACS analysis. (B) The bar graph is shown in panel A. (C) The expression of caspase 3, caspase 9, and PARP in the cells was analyzed via western blotting. GAPDH was used as a loading control. (D, E, and F) The cells were pretreated with MitoTEMPO for 1 h and then treated with IQ for 18 h. (D) The cells were analyzed via MTT assay. (E) The cells were examined via annexin V-PI analysis. (F) The bar graph is shown in panel E. The experiments of penal B, D, and F were performed at least three times, and data shown as the mean ± SD. Statistical analyses were performed using Student’s t-test. **, p < 0.05 and ***, p < 0.001 compared with the control.
Comment: Figure 5F. p-value in legend does not appear to be consistent with ‘***’ on graph, please check and correct
Answer: According to Reviewer’s comment, we check again the p-value in Figures 5F. The p-value was 0.000984837, which is less than 0.001. Thus, the legend of Figure 5F was corrected as follow.
The experiments of penal B, D, and F were performed at least three times, and data shown as the mean ± SD. Statistical analyses were performed using Student’s t-test. **, p < 0.05 and ***, p < 0.001 compared with the control.
Comment: References. Reviewer found several entries where capitalization/punctuation was not consistent throughout listings. Please check carefully.
Answer: Thanks for Reviewer’s comment. We used Endnote for organizing references. However, we did not carefully check the References before submission. We have checked the Reference again by manually in this revision.
Round 2
Reviewer 1 Report
Most of my concerns are adequately addressed. I support acceptance.